# Flaxseed Powder Attenuates Non-Alcoholic Steatohepatitis via Modulation of Gut Microbiota and Bile Acid Metabolism through Gut–Liver Axis

**DOI:** 10.3390/ijms221910858

**Published:** 2021-10-08

**Authors:** Chao Yang, Min Wan, Dengfeng Xu, Da Pan, Hui Xia, Ligang Yang, Guiju Sun

**Affiliations:** 1Key Laboratory of Environmental Medicine and Engineering of Ministry of Education, School of Public Health, Southeast University, Nanjing 210009, China; wenzhengwuguan@yeah.net (C.Y.); WMlove92YC@126.com (M.W.); withxu@seu.edu.cn (D.X.); pantianqi92@foxmail.com (D.P.); huixia@seu.edu.cn (H.X.); 2Department of Nutrition and Food Hygiene, School of Public Health, Southeast University, Nanjing 210009, China

**Keywords:** flaxseed powder, non-alcoholic steatohepatitis, bile acid, gut microbiota, FXR, TGR5

## Abstract

Non-alcoholic steatohepatitis (NASH) is gradually becoming one of the most common and health-endangering diseases; therefore, it is very important to prevent the occurrence of NASH and prevent simple non-alcoholic fatty liver (NAFL) from further developing into NASH. We fed mice a high-fat diet (HFD, 60% fat) for 14 weeks to induce NAFL and then fed different doses of flaxseed powder (low (10%), middle (20%), and high (30%)) to the mice for 28 weeks. After the animal experiment, we analyzed fecal bile acid (BA) profiles of the HFD mice, flaxseed-fed (FLA-fed) mice, and control mice with a normal diet (10% fat) using a targeted metabolomics approach, and we analyzed the gut microbiota at the same time. We also investigated the mechanistic role of BAs in NASH and identified whether the altered BAs strongly bind to colonic FXR or TGR5. In the present study, we found that 28-week FLA treatment notably alleviated NASH development in NAFL model mice fed with an HFD, and the beneficial effects may be attributed to the regulation of and improvement in the gut flora- and microbiota-related BAs, which then activate the intestinal FXR-FGF15 and TGR5-NF-κB pathways. Our data indicate that FLA might be a promising functional food for preventing NASH through regulating microbiomes and BAs.

## 1. Introduction

It is estimated that about 25% of the global population have non-alcoholic fatty liver disease (NAFLD) [1]. Non-alcoholic steatohepatitis (NASH) is characterized by hepatic steatosis and inflammatory cell infiltration, which is the progressive form of NAFLD [2]. The incidence of NASH is projected to increase by up to 56% in the next 10 years [3]. Compared to simple non-alcoholic fatty liver (NAFL), the pathological process of NASH is irreversible, and it has the potential to develop into cirrhosis and hepatocellular carcinoma (HCC) [4]. Therefore, it is highly necessary to prevent the progression of NAFL into NASH.

A growing body of experimental and clinical evidence suggests that the gut microbiota is related to NAFLD pathogenesis [5]. Recent studies have suggested that bile acids (BAs), as metabolites of gut microbiomes [6,7], may have unique therapeutic promise for NASH. BAs are molecules synthesized from cholesterol in the liver and then secreted to the gut where they can be further metabolized by the microbiota [8]. In general, BAs are synthesized in the liver via two different routes: The classical pathway is initiated by the rate-limiting enzyme CYP7A1, and it converts cholesterol to cholic acid (CA) and chenodeoxycholic acid (CDCA). The CYP8B1 enzyme is required for CA synthesis in the classical pathway. The alternative pathway is initiated by CYP27A1 and followed by CYP7B1 to generate CDCA [9]. The alternative pathway predominantly generates CDCA, while the classical pathway generates both CDCA and CA [10]. CYP8B1 activity determines the ratio of CA to CDCA [10]. The structural difference between CA and CDCA is that CA and its derivatives have a 12α-OH that is catalyzed by CYP8B1, while CDCA and its derivatives are non-12α-OH BAs without 12-hydroxylation [11].

Of note, the primary bile acids produced in humans are CDCA and CA, while rodents produce CA and muricholic acids (MCAs), predominantly beta-MCA (βMCA) [10]. In humans, ursodeoxycholic acid (UDCA) is a secondary bile acid, and MCAs are generally not detected. It has been suggested that the human gut microbiota is unable to metabolize βMCA [12], which suggests warranting caution when translating findings from mice to humans. In addition, bile acids are amidated with the amino acid glycine and a small amount of in humans. In contrast, bile acids are almost exclusively conjugated with taurine in mice and rats [13].

Of the two pathways, the classical pathway is gradually being recognized as a key player in regulating lipid, cholesterol, carbohydrate, and energy homeostasis [13]. Additionally, increases in the ratios of 12αOH BAs result in their loss of ability to control lipid homeostasis and inflammatory conditions [13,14]. An increase in 12α-OH BA levels in NASH patients has been identified [6]. Recent studies have also revealed that higher plasma 12αOH/non-12αOH BA ratios were related to lower insulin sensitivity in humans [15] and that fecal 12αOH Bas were strongly associated with hepatic steatosis [16]. Therefore, regulating BA metabolism might be an effective strategy to prevent NAFLD and related disorders [17]. Several similar studies showed that BAs can be considered as potential therapeutic targets in NAFLD [18,19].

Studies in humans have demonstrated that the gut microbiota is altered in obesity and related diseases, and mouse studies have demonstrated that the gut microbiota contributes to disease phenotypes [20]. Gut bacteria-derived bile salt hydrolases (BSHs) are major enzymes that catalyze the “gateway” reaction in the bacterial metabolism of BAs [21]. Studies regarding BSHs related to hosting health are primarily performed using mouse models. *Firmicutes*, *Proteobacteria*, and *Actinobacteria,* known phyla that harbor bacteria with high BSH activity, had positive correlations with BAs [22]. In the colon, primary BAs synthesized by hepatocytes can be transformed into secondary BAs through the action of these certain microbiomes and re-enter the enterohepatic circulation of BAs [10,23]. Changes in BAs can induce the ileum to secrete fibroblast growth factor-15 (FGF15, human FGF19). When FGF15 is transported to the liver, it inhibits Cyp7a1 transcription and further limits the synthesis of BAs [24]. In addition, given the signaling properties of BAs, changes in BAs could also activate the farnesoid X receptor (FXR), and the membrane Takeda G-protein coupled receptor 5 (TGR5) [25,26,27], in order to regulate lipid metabolism and inflammation. This close metabolic pathway is called the “gut–liver axis”. Additionally, gut–liver axis alteration can lead to NAFLD development and might favor the occurrence of NASH [28].

Currently, although there is an urgent need for the management of NASH, no approved pharmacologic therapy is available. Nutritional intervention is one of the important choices for improving NASH. Flaxseed (*Linum usitatissimum* L.), due to its functional components, can be considered as a promising adjunct intervention for reducing inflammation and hepatic steatosis in NAFLD patients [29,30]. Existing studies have shown that the main bioactive components of flaxseed such as flaxseed polysaccharide [31], flaxseed oil [32,33,34,35], and flaxseed lignan [36] had an effect on inhibiting inflammation and improving lipid metabolism. Additionally, its effects of modulating the gut microbiota have been found [37,38,39]. Due to the close relation between microbiomes and BA conversion and metabolism, the interaction between the gut microbiota with BAs induces great concern in studying BA homeostasis, lipid metabolism, and metabolic inflammation.

In this study, we fed mice a high-fat diet (HFD, 60% fat) for 14 weeks to induce simple hepatic steatosis and then fed different doses of flaxseed powder to the mice for 28 weeks. We analyzed fecal BA profiles of the HFD mice, flaxseed-fed (FLA-fed) mice, and control mice with a normal diet (10% fat) using a targeted metabolomics approach in the present study. Among the five groups of mice, we identified distinct fecal BA profiles which were presumably derived from different gut microbiota compositions. Therefore, we analyzed the gut microbiota in control, HFD, and three FLA-fed groups. We also investigated the mechanistic role of BAs in NASH in which specific BAs strongly bound to colonic FXR or TGR5 and sustained activation of downstream liver FGF15 and intestinal NF-κB p65, leading to significantly improved hepatic BA homeostasis, lipid metabolism, and metabolic inflammation.

## 2. Results

### 2.1. The Different Physiological Changes and Biochemical Parameters in the Control, HFD and FLA-Fed Groups

The therapeutic effect of FLA was focused on in the present study, where we conducted experiments in which we administered NAFL mice FLA to verify the delaying NASH effects of flaxseed powder. Flaxseed powder intake reduced weight gain, hyperlipidemia, and inflammation that the HFD induced. Compared with the HFD, the three FLA-fed groups exhibited more gradual weight gain (Figure 1A) throughout the whole experimental period. At the end of the 28 weeks of the interventional phase, the final body weights of the three FLA-fed groups were significantly lower than the HFD group (*p* < 0.05), in which the high-dose FLA-fed group was reduced gradually to the level of the control group (Figure 1B). In addition, the daily fat intake was not different between HFD and FLA-fed mice (Figure 1C).

The results show that the serum level of TG, LDL-C, ALT, and TNF-α was decreased by the FLA-fed intervention (Figure 2B,C,E,H), and the HDL-C concentration increased in the middle dose of flaxseed powder (Figure 2D). Therefore, our results indicate that flaxseed powder consumption for mice could markedly prevent HFD-induced weight gain and decrease hyperlipidemia and inflammation. Moreover, the serum level of TC, AST and IL-6 in FLA-fed groups had reduction with non-statistical difference, compared with HFD group (Figure 2A,F,G). 

Histological NAS assessment was performed on HE-stained hepatic tissue. There was no obvious steatosis, inflammation, and ballooning degeneration observed in the control group (Figure 3A). In contrast, macro-vesicular steatosis, with inflammation and ballooning degeneration, developed in NASH mice. The results from the NAS scoring system show that mice in the HFD group had significantly higher (*p* < 0.05) scores for all categories than those in the control group (Figure 3B,F–H). Mice in the FLA-fed intervention groups (middle and high) had markedly decreased (*p* < 0.05) NAS scores compared to the HFD group (Figure 3C–H).

### 2.2. Disorders of BA Profiles in the HFD Group and a Relative Increase in the Proportion of Non-12αOH BAs in the FLA-Fed Groups

A clear separation was observed between the control, HFD, and flaxseed intervention group (middle and high doses) mice using the sparse partial least squares discriminant analysis (sPLS-DA) model that was established by the detected fecal BAs (Figure 4A). The PCA score plot is shown in Figure 4B. A heat map displaying the means for each type of fecal BA in each study group is shown in Figure 4C. Similarly, a trend of separation between the control, HFD, and flaxseed intervention groups was also observed. The variable importance in projection (VIP) scores from the sPLS-DA model indicated that non-12-OH acids were the top BAs that resulted from the group separation between the HFD and FLA-fed groups (Figure 4D).

Significant changes in the ratios of non-12αOH BAs, 12αOH BAs, or non-12αOH/12αOH between control, HFD, and FLA-fed groups, in which the ratios of non-12αOH BAs in feces were significantly increased in the middle- and high-dose FLA-fed groups compared with the HFD group (Figure 5A), while 12αOH BAs showed no significant differences (Figure 5B). The trend of the proportion of non-12αOH/12αOH was similar to non-12αOH (Figure 5C).

We quantitatively analyzed the concentration of BAs in the feces of mice fed with control chow and the HFD and found that the HFD significantly increased fecal BA levels (Appendix A). Compared to the control group, 29 of the 41 quantified BAs in the HFD group had higher concentrations. All the CA-derived BAs, including TDCA, TCA, and DCA, were significantly increased, indicating that 12αOH BAs might be the main BA spectrums that responded to the HFD (Figure 6A,B and Appendix A).

Compared to the HFD intervention, mice fed with the middle dose of flaxseed significantly increased UDCA and HDCA, which belonged to non-12α BAs (Figure 6C). Moreover, among the non-12α BAs, seven had higher concentrations in the high-dose flaxseed-fed group, including UDCA, HDCA, LCA, alpha-MCA, beta-MCA, glycohyodeoxycholic acid (GHDCA), taurohyodeoxycholic acid (THDCA), and TUDCA, with statistical significance (Figure 6D and Appendix A). However, we also found that two BAs, DCA and norcholic acid (NorCA), were significantly increased after a high-dose flaxseed intervention.

### 2.3. Gut Microbiome Alteration Induced by FLA-Fed and Its Correlation with BAs

The effect of FLA on the fecal microbiota composition was detected by sequencing the respective 16S rRNA genes. The gut microbiota of eight mice in each group was analyzed. Amplicon sequence variants/operational taxonomic units (ASV/OTUs) were detected. Among these, only 159 were shared by all groups, accounting for 0.83%. There were 4287, 2717, 2971, 2238, and 4865 unique ASV/OTUs in the Co, HFD, and three FLA-fed groups (from low to high) (Appendix A). The rarefaction curve, Shannon index curve, rank abundance curve, and species accumulation curve were calculated (Appendix A). Alpha diversity (including Chao1, observed species, Pielou’s evenness, Simpson and Shannon indices) was found to be unchanged; however, there were significant differences in evenness and coverage as Faith’s phylogenetic diversity (Faith’s PD) index and Good’s coverage index show (Figure 7A). The results show that there were phylogenetic differences between species among five dietary groups by Faith’s PD index, and differences between the percent of an entire species by Good’s coverage index. The plot of the orthogonal partial least squares discriminant analysis (OPLS-DA) at the genus level shows that the gut microbiota composition was significantly different among the five groups (Figure 7B). The OPLS-DA between the control, HFD, and FLA-fed groups indicated that the genera *Bifidobacterium*, *Allobaculum*, *Desulfovibrio*, *Oscillospira*, *Bacteroides*, *Adlercreutzia*, *Mucispirillum*, *X. Ruminococcus*, *Clostridium*, and *Turicibacter* were the top 10 gut microbiota that resulted in a separation between the control, HFD, and FLA-fed groups (Figure 7B).

To reveal the degree to which the gut microbiota of the FLA-fed groups differed from that in the control and HFD groups, the intestinal microbiota structural changes were subsequently analyzed by using PCA, PCOA, and NMDS based on unweighted Unifrac or Jaccard (Figure 8). All of them indicated that the flaxseed diets led to a distinguishable microbiome composition compared with the HFD, which resulted in a similar microbiome composition to the control diet (Figure 8A–C). The differences in the bacterial taxa composition (Figure 8B) between different groups were observed using an unweighted PCOA. The PCOA plot revealed a significant separation between the five different groups along the PC1 axis (9.6% of total variation). Additionally, 8.7% of the variability in microbial communities was explained by the PC2 axis. Additionally, there are significant differences among the five groups (with R = 0.25, *p* = 0.001), using Adonis methods. Consistently, UPGMA analysis showed that the samples in the control, FLA-fed, and HFD groups showed hierarchical clustering based on the Jaccard similarity (Appendix A).

Population analyses were performed, and the mean percentage of the total population at six levels for each diet group is shown in Appendix A. To compare the gut microbiota between the control, HFD, and FLA-fed groups, the linear discriminant analysis effect size (Lefse) method was also performed. This method presents the cladograms from kingdom to genus at six different levels (Appendix A) and shows the biomarker taxa linear discriminant analysis (LDA) scores of > 2, which were derived from Lefse analysis (Appendix A). In the control group, the phyla *Firmicutes*, *Bacteroidetes,* and *Proteobacteria* had a large effect size. In the HFD group, the classes *Clostridia*, *Deltaproteobacteria*, and *TM7-3* had a large effect size. In the FLA-fed groups, the phyla *Firmicutes*, *Bacteroidetes*, and *Proteobacteria* had a large effect size (Figure 9A).

The changes in the relative abundance of *Firmicutes* and *Bacteroidetes* have been regarded as the two predominant bacterial divisions that alter the genetic composition and metabolic activity of the mammalian gut microbiome [40]. In the FLA-fed groups, the relative abundance of the phylum *Firmicutes* decreased while that of the phyla *Bacteroidetes* and *Actinobacteria* increased (as shown in Figure 9). The analysis of the gut microbiota showed that the phylum levels of *Firmicutes* and *Actinobacteria* had high BSH activity [41], although *Bacteroidetes*, a major bacterial phylum with low BSH activity, was altered after FLA treatment. In the phylum *Firmicutes*, the relative abundances of several families, including *Lachnospiraceae* and *Clostridiales*, increased in response to the HFD, while those of the genus *Lactobacillus* decreased, compared to the control diet. The abundance of the families *Lachnospiraceae* and *Clostridiales* decreased obviously in the FLA-fed mice. In the phylum *Bacteroidetes*, the abundance of the families *S24-7* and *Rikenellaceae*, the most abundant families in the control group, decreased markedly in the HFD mice (as shown in Appendix A). However, compared to the HFD group, the relative abundances of the families *S24-7* and *Rikenellaceae* increased markedly in the FLA-fed mice.

In addition, the visual differences in the bacterial compositions at the genus level are presented using heatmap analysis in Appendix A. For the compositions at the genus level, the FLA-fed groups and control group were more similar to each other, whereas the FLA-fed groups were extremely different from the HFD group.

To visualize the correlation of the gut microbiota and BA abundances, Spearman correlation was conducted between the main non-12α BA abundances in feces and the relative abundances of the differential bacteria species identified above. We found that the phylum *Bacteroidetes*, especially the family *S24-7,* had significantly positive correlations with HDCA, LCA, UDCA, GHDCA, α-MCA, and β-MCA. The phylum *Firmicutes* had a significant negative correlation with HDCA, LCA, UDCA, GHDCA, α-MCA, and β-MCA (Figure 10). These results suggest that gut microbiota changes may impose a substantial impact on the BA composition.

### 2.4. FLA-Fed Alleviates NASH via Modulating BA Metabolism by FXR/FGF15 and TGR5/NF-κB Signaling

We further performed mRNA analysis of liver tissue to evaluate the expression of enzymes responsible for BA synthesis, in order to clarify the reason and mechanisms of altering BA profiles induced by FLA. Whereas liver FXR gene expression was not significantly changed, liver Cyp7a1 and Cyp8b1 expressions were significantly increased in the HFD-fed group compared with the control group. Meanwhile, liver Cyp7a1 and Cyp8b1 expressions in the FLA-fed groups were decreased compared to the HFD group (Appendix A). Consistently, the Western blot results show that the expression of the CYP7A1 protein was reduced in the FLA-fed groups (Figure 11B).

BA synthesis is regulated not only by the liver FXR/SHP pathway but also by the intestinal FXR/FGF15 signal. Findings have revealed that CYP7A1 was regulated more strongly by the intestinal FXR/FGF15 pathway [42,43]. We detected the FXR protein in the ileum, which was significantly downregulated in the HFD group compared with the control group; however, the FLA-fed groups significantly increased FXR protein expression compared with the HFD group (Figure 11E). Moreover, we found that there were differences in the plasma FGF15 concentration in the FLA-fed groups compared to the HFD group (Appendix A). In addition, we examined the expression of FGFR4 of the intestinal FXR pathway and found that the FGFR4 protein was significantly overexpressed in the liver tissue of the HFD group mice, whereas there were significant decreases in FGFR4 expression in the FLA-fed groups (Figure 11C). Activation of FXR downregulates Cyp8b1 expression by upregulating SHP and FGF15/19 levels [44]. However, hepatic FXR signaling mainly suppresses Cyp8b1 expression, and intestinal FXR signaling is functional for downregulating both Cyp7a1 and Cyp8b1 expressions [44]. Cyp8b1 is the product that is crucial for catalyzing hydroxylation at position 12 in hepatic CA synthesis [45].

TGR5 is another nuclear receptor for bile acid, which is crucial for metabolic disorders and liver inflammatory diseases. Analysis of intestinal protein expression revealed differences in the expression levels of TGR5 among five groups. HFD feeding markedly suppressed the expression of TGR5 compared with the control group. By contrast, FLA supplementation increased the expression of TGR5 compared with the HFD group (Figure 11F). To explore the molecular mechanism underlying the improvement in FLA-induced liver inflammation, we further investigated the hepatic expression levels of proteins involved in inflammatory metabolism. The results show that the expression level of NF-κB P65 and TLR4 decreased in the FLA-fed groups compared with the HFD diet (Figure 11A,D).

## 3. Discussion

NASH is gradually becoming one of the most common and health-endangering diseases; therefore, it is very important to prevent the occurrence of NASH and prevent NAFL from further developing into NASH. In the present study, we found that 28-week FLA treatment notably alleviated NASH development in NAFL model mice fed with an HFD, and the beneficial effects may be attributed to the regulation of and improvement in the gut flora and microbiota related BAs, which then activate the intestinal FXR-FGF15 and TGR5-NF-κB pathways. Our data indicate that FLA might be a promising functional food for preventing NASH through regulating microbiomes and BAs.

There is the largest number of bacteria and species in the gut. Of the gut microbiota, *Firmicutes* and *Bacteroidetes* are the dominating microbiotas, and *Firmicutes*/*Bacteroidetes* (F/B) ratio rises in obese and NASH patients have been reported [46,47]. Diet intervention can reduce the ratio of F/B and improve intestinal dysbiosis and metabolic disorders [48]. We observed that the effects of FLA treatments can be attributed to the modulation of gut dysbiosis in NASH mice. In the FLA-fed groups, the relative abundance of the phylum *Firmicutes* decreased, while the abundance of the phyla *Bacteroidetes* and *Actinobacteria* increased, compared to the HFD group. Gut microbiota analysis revealed that the phyla *Bacteroidetes*, *Firmicutes*, and *Actinobacteria* had high BSH activity [41,49]. We reveal that FLA treatment altered these gut microbiotas with an increase in the intestinal flora in BSH-containing phyla. Islam et al. [50] first observed that CA increases in the *Firmicutes* phylum and decreases in the *Bacteroidetes* phylum in rats; this microbial signature was just the opposite to that of the FLA-fed gut microbiome in our study.

Studies found that bacteria including *Lactobacilli*, *Bifidobacterium*, *Clostridium*, and *Bacteroides* can deconjugate BAs [41] in which the functional BSH is present [10]. A previous study showed that alteration in BA hydrophobicity through BSH deconjugating induced the expression of key genes related to lipid metabolism and transportation [51]. Supporting evidence showed that the abundances of *Lactobacilli* and *Bifidobacterium* had positive correlations with the secondary/primary BA ratio [52], and the abundance of *Firmicutes* was negatively associated with the secondary/primary BA ratio. Our results are consistent with the above-mentioned results that the abundance of *Bifidobacterium* increased in FLA-fed groups (middle and high dose) and *Firmicutes* decreased in all FLA-fed groups. Moreover, we observed that the FLA-fed groups increased the abundance of *Clostridium* (cluster XI and XIVa) compared to the HFD group. *Clostridium* (clusters XIVa and XI), which belongs to *Firmicutes*, is responsible for the change from primary to secondary BAs, which is in charge of 7α-dehydroxylation to specifically generate DCA in the large intestine [53]. In addition, hydroxysteroid dehydrogenases (HSDHs), which **are** responsible for another major biotransformation of BAs to generate oxo- (or keto-) BAs, are present in *Actinobacteria*, *Proteobacteria*, *Firmicutes*, and *Bacteroidetes* [10].

Recently, a study suggested the 12αOH-to-non-12αOH BA ratio was extremely high in NASH patients [54]. The increased secretion of 12αOH BA may lead to hepatic steatosis by promoting fat absorption and liver Cidea mRNA expression [16]. The increase in 12αOH BAs had an association with enhancements of liver Cyp7a1 and Cyp8b1 mRNA expression. Of the two enzymes, Cyp7a1 is the rate-limiting enzyme of BA synthesis [9], and Cyp8b1 is responsible for producing 12αOH BAs [10]. A study found that the HFD increased liver Cyp7a1 and Cyp8b1 expression, which preferentially increased liver 12αOH BA synthesis by using accumulated hepatic Chol [17]. In the present study, our results support the idea that Cyp7a1 and Cyp8b1 expression was significantly increased in the HFD group compared to the control group.

Among the non-12αOH BAs, seven had higher concentrations in the high-dose flaxseed-fed group, including UDCA, HDCA, LCA, alpha-MCA, beta-MCA, GHDCA, THDCA, and TUDCA, compared to the HFD group. However, we also found that two 12αOH BAs, DCA and NorCA, were significantly increased. Evidence has shown that DCA and LCA, as secondary Bas, were more effective in activating TGR5 than primary BAs (CA and CDCA) [11,55]. Therefore, the gut microbe-dependent production of DCA, as well as its enterohepatic circulation and conjugation in the liver, is of profound significance in liver fibrogenesis. Moreover, the increase in UDCA protected the liver from steatosis [56]. UDCA has been proved to be effective in the treatment of various liver diseases, including NASH [57]. UDCA was found which not only had an enhancement on GLP-1 release in vitro but also had a mitigating effect on HFD-induced obesity by oral administration [58]. Moreover, a mixture of UDCA and LCA showed a reduction in hyperlipidemia and could alleviate obesity, hyperglycemia, and hepatic steatosis in HFD-fed mice by treating them with a bacterial strain capable of elevating UDCA, LCA, and succinate levels [59].

As signal molecules, BAs not only regulate their biosynthesis but also regulate key metabolic pathways by activating FXR and TGR5 [60,61]. As FXR and TGR5 are known to play key roles in regulating BA synthesis and homeostasis, we therefore investigated whether the effects of the intestinal microbiota improved by FLA on BA synthesis are mediated through FXR and TGR5. In the present study, we firstly revealed that FLA activated intestinal the FXR-FGF15 signaling pathway to modulate hepatic BA disorders. We analyzed the expression of FXR in the ileum and found that the expression of intestinal FXR was upregulated after flaxseed feeding. Consistently, there was an increase in the plasma FGF15 concentration in the FLA-fed groups compared to the HFD group. These results **prove** that FXR is widely expressed in the intestine and tightly controls enterohepatic BA homeostasis and hepatic improvement. There was a report that metabolic disorder was partly improved through stimulating intestinal FGF15 production by intestine-restricted FXR agonist fexaramine [62]. In addition, intestinal FXR activation can resist an LPS-induced damaged intestinal epithelial barrier [63], which may be associated with reducing translocation of bacteria to the liver in cirrhosis [64]. Recent studies also showed that intestine-derived FGF15 regulates hepatic BA synthesis [10]. To further explore the expression of downstream signal proteins or genes of FXR-FGF15, liver CYP7A1 protein and Cyp8b1 expression were detected, and it was found that both decreased in the FLA-fed groups when compared with the HFD group. Evidence has shown overexpression of CYP8B1 and CYP7A1 in HFD-induced NAFLD [65]. Emerging evidence has suggested that inhibition of intestinal FXR-FGF15 could lead to the improvement in NAFLD, obesity, and hypercholesterolemia [22,66,67]. Among them, CYP7A1 is the enzyme for BA synthesis rate limiting, whereas CYP8B1 is the enzyme determining the ratio of CA/CDCA [10]. We found that FLA-fed mice reduced CYP8B1 significantly, thereby resulting in a relatively expansionary proportion of the non-12αOH BAs in the FLA-fed group. Concerning the regulation of BA metabolism, Cyp8b1^-/-^ in mice specifically reduced the 12αOH BAs, which protected HFD-induced hepatic steatosis due to damaged fat absorption [68]. Additionally, CYP7A1 is more strongly regulated by intestinal FXR, while CYP8B1 is more sensitive to liver FXR activation [42]. Intestinal microbiota and BA metabolism are in mutual regulation. Changes in BA signatures, as well as modulation of the gut–liver FXR/FGF15/CYP7A1 axis, were found in aseptic, probiotic-, and antibiotic-treated animals [62]. In addition, for FGFR4 in the FXR-FGF15 pathway, as another key receptor, our results show that flaxseed intervention significantly reduced the expression of FGFR4. Upregulated expression of FGFR4 was recently demonstrated in murine NASH models [69] and patients developing HCC related to fatty liver disease [70,71]. A study found FGFR4 could be specifically inhibited by activating a soluble FGFR4 extracellular domain fragment in vitro, and that it suppressed steatosis and the development of fatty liver in mice [72].

TGR5 and BA metabolism have been important targets for pharmacologic approaches. With regard to TGR5, it has implications in modulating inflammatory conditions. It has been reported that LPS-induced inhibition of Caco-2 cell proliferation could be reversed by TGR5 overexpression, whereas the opposite effect was found by knockdown of TGR5 [73]. In our present study, we found that FLA supplementation increased the expression of TGR5 compared with the HFD group, and the expression level of NF-κB P65 and TLR4 decreased in the FLA-fed groups compared with the HFD. TGR5 activation appears to induce potent anti-inflammatory effects through inhibiting the nuclear translocation of NF-κB and suppressing cytokine production [74,75].

There are several limitations in this study. First, we did not measure BAs in the serum blood and liver tissue, in which FLA may also affect BAs and BA receptors. However, the bile acid profile of feces reflected the profiles in the cecum and colon [26], which could better reflect the role of the intestinal flora. Second, although we observed changes in BAs which led to activation or inhabitation of FXR and TGR5, the decreased activity of two BA receptors during the HFD might be the result of a low gut BA level, and the increase in two receptors’ activities by FLA feeding might be the result of direct stimulation of these two receptors with unknown metabolites and mechanisms. Third, if we could add control groups on a standard diet supplemented with flaxseed powder, we will better clarify the role of flaxseed powder itself in regulating the intestinal flora and bile acids.

## 4. Materials and Methods

### 4.1. Animal Experiments and Diets

The animal experiment was approved by the Institutional Animal Care and Use Committee of Southeast University (No. 2019-0725-016), and the animal experimental protocols were conducted according to the “Principles of Laboratory Care”. Male, six-week-old mice (C57BL/6, specific pathogen-free grade (SPF)) were purchased from Shanghai Lingchang Biotechnology Co., Ltd. and were allowed one-week acclimatization by placing them on a control chow diet. All experimental mice were kept in SPF environments, which are under a controlled condition of a 12 h light/12 h dark cycle at 21–23 °C and 40  ±  5% humidity. Mice had free access to chow and ultrapure water.

In the flaxseed powder intervention study, mice were firstly divided into two groups for 14 weeks: (1) normal diet group (n = 8): mice were fed chow diets consisting of 10% fat, 71% carbohydrate, and 19% protein; (2) high-fat diet (HFD) group (n = 42): mice were fed with high-fat diets: 60% fat, 20% carbohydrate, and 20% protein. All chows were produced by Jiangsu Xietong pharmaceutical bio-engineering Co..LTD (Nanjing, China). Flaxseed powder was purchased from CanMar foods Ltd., Regina, Canada. The ingredients of the flaxseed powder used are shown in Appendix A. Every four mice were housed in one cage, and food intake and body weight were measured once a week during the whole experiment.

Then, 10 mice in the HFD group were dissected, and their livers were taken for pathology to confirm the establishment of a simple steatosis model by the scoring system of the Kleiner NAFLD activity score (NAS) [76] (NOS score is 2.88 ± 0.52). Subsequently, these simple steatosis mice were randomly divided into four groups (eight mice per group): an HFD group that received a high-fat diet, and low (10%)-, middle (20%)-, and high (30%)-dose flaxseed powder groups that received a high-fat diet containing 10%, 20%, and 30% flaxseed powder (the timeline showing the experimental design is shown in Appendix A). Flaxseed powder was added to the HFD according to certain proportions. The total energy, protein, carbohydrate, and fat among the four groups were balanced. The details of the feed formulations for each group are shown in Appendix A.

The body weights and food intake of all animals were recorded once a week during the experiments for the next 28 weeks. Livers, epididymal, abdomen, and kidney white adipose tissues, brown adipose tissues, and cecal content were stored in −80 °C refrigerators after careful dissection and snap freezing in liquid nitrogen.

### 4.2. Serum Sample Collection and Analysis

At the end of the experiment, mice were fasted for 12 h and then anesthetized for blood collection. Serum blood was harvested at 4 °C, by centrifuging the blood at 3700× *g* for 25 min. Serum total cholesterol (TC), low-density lipoprotein cholesterol (LDL-C), high-density lipoprotein cholesterol (HDL-C), triglyceride (TG), alanine aminotransferase (ALT), and aspartate aminotransferase (AST) were measured by enzymatic methods using commercial assay kits (Jiancheng Co., LTD, Nanjing, China). Serum interleukin-6 (IL-6), tumor necrosis factor-α (TNF-α), and interleukin-6 (IL-6) were determined using commercial ELISA (Enzyme-Linked ImmunoSorbent Assay) Kits (Nanjing Jin Yibai Biological Technology Co. Ltd., Nanjing, China).

### 4.3. Liver Histological Analyzes

Tissues were fixed in formalin, then embedded in paraffin, and tissue sections were stained with hematoxylin and eosin (HE). The diagnosis and staging of NASH were established based on the scoring system of the Kleiner NAFLD activity score (NAS) [76]. In summary, the pathological images were obtained by an optical microscope (Olympus BX41) at 400× magnifications. The results were analyzed by an external pathology researcher and were based on the degree of steatosis, focal and diffuse inflammation, and nuclear vacuolation. According to the NAS score at liver biopsy, NAFLD was classified as non-NASH (NAS < 3), borderline NASH (NAS 3-4), and NASH (NAS > 5).

### 4.4. Quantitative Real-Time PCR

The liver and intestinal tissues were collected and stored in StarRNA Tissue Stabilizer (Genstar, Beijing, China) immediately for 12 h and then stored at −80 ◦C for further experiments. Firstly, total RNA was extracted using TRIzol reagent (Ambion, CA, USA), and then RNA was converted to complementary DNA (cDNA) using Fist-strand cDNA Synthesis Kits (Genstar, Beijing, China). RT-qPCR was performed on the CFX Connect Real-time System (BIO-RAD, Singapore), and the thermal cycle conditions were described in Liu et al. [77]. Gene-specific primers are the listed sequences in Appendix A. Targeted gene levels were normalized to glyceraldehyde-3-phosphate dehydrogenase (GAPDH) gene levels, and the results were analyzed by using the ΔΔCT analysis method [78].

### 4.5. Western Blot Analyses

Liver and intestine tissues were lysed in cold RIPA buffer with protease inhibitors (YEASEN, Shanghai, China). The concentrations of the protein were quantified with the BCA protein assay kit (ThermoFisher, 23227). Equal amounts of protein (100 μg per lane) were electrophoresed on sodium dodecyl sulfate (SDS, 10%)-polyacrylamide gel electrophoresis and transferred to a PVDF membrane. The membranes were blocked at room temperature with 5% non-fat milk powder, and the membranes were incubated with primary antibodies at 4 °C overnight. Primary antibodies used were FXR (1:1000, mAb #72105, Cell Signalling Technology, Beverly, MA), TGR5 (ab72608, Abcam, Cambridge, United Kingdom), CYP7A1 (1:1000, ab138497, RRID: AB_2828001, Abcam, Cambridge, United Kingdom), FGFR4 (1:1000, mAb#A9197, ABclonal, Wuhan, China), TLR4 (1:1000, sc-293072, Santacruz, USA), and NF-κB p65 (1:1000, mAb #8242, Cell Signalling Technology, Beverly, MA). The relative intensity of bands was normalized to β-actin using Beta Actin antibody (1:1000, Cell Signalling Technology, Beverly, MA). The membranes were washed with TBST buffer three times and followed by incubating at room temperature with HRP-conjugated secondary antibody for 2 h (1: 20000, SA00001-1/SA00001-2, Proteintech Group, Inc., USA). ImageJ software was used to quantify the gray values of gel bands.

### 4.6. Gut Microbiota Assessment

All eight mice from each group were selected for gut microbiota analysis. Community DNA was extracted from the cecal contents collected from the C57BL/6 mice at the end of the 28-week feeding experiment. Total genome DNA was extracted from mouse colonic feces using the CTAB/SDS method. The V3–V4 variable regions of the bacterial 16S rRNA genes were amplified. All DNA samples were amplified and purified using a primer pair (338F and 806R). PCR products were mixed in equidensity ratios. Then, the mixture of PCR products was purified with Qiagen Gel Extraction Kit (Qiagen, Germany). Finally, high-throughput sequencing was performed on the Illumina NovaSeq platform (Illumina, USA). A quality-checked DNA sample was used to construct metagenome shotgun sequencing libraries by using the Illumina TruSeq Nano DNA LT Library Preparation Kit. The original data obtained by sequencing were spliced, assembled, and filtered, and chimeras were removed to obtain the effective data (details in Appendix B and Appendix C). Operational taxonomic unit (OTU) clustering was conducted by QIIME2 (2019.4) software and delimited at 95% of the threshold using qiime feature-table rarefy. According to the results of OTU clustering, species annotation was conducted for the representative sequences of each OTU to obtain the corresponding species information and species-based abundance distribution. Alpha diversity was calculated by QIIME2 (http://scikit-bio.org/docs/latest/generated/skbio.diversity.alpha.html#module-skbio.diversity.alpha, 1 March 2021), including Chao1, observed species, Shannon index, Faith’s PD, Pielou’s evenness, and Good’s coverage index, to assess the community diversity of microbes in the intestinal flora. Rarefaction curves, species accumulation curves, and rank abundance curves were also drawn. Principal component analysis (PCA), principal coordinate analysis (PCoA), nonmetric multidimensional scaling (NMDS), and the unweighted pair-group method with arithmetic means (UPGMA) were used to evaluate beta diversity based on unweighted Unifrac distance. Significant differences in beta diversity were analyzed by analysis of Venn (https://en.wikipedia.org/wiki/Venn_diagram, 1 March 2021). Linear discriminant analysis effect size measurement (Lefse) analysis was performed using the Galaxy online platform (http://huttenhower.sph.harvard.edu/galaxy/, 1 March 2021). Orthogonal partial least squares discriminant analysis (OPLS-DA) was conducted to display differences in species abundance composition between groups. The highest number of top 100 species was used to image heat maps by the R Programming Language.

### 4.7. Bas Analysis

Concentrations of fecal Bas were measured according to previously reported methods [11]. In short, the Waters ACQUITY ultra-performance LC system and Waters XEVO TQ-S mass spectrometer with an ESI source controlled by MassLynx 4.1 software (Waters, Milford, MA) were used for analyzing fecal extracts and BA reference standards. An ACQUITY BEH C18 column (1.7 µm, 2.1 × 100 mm internal dimensions) (Waters, Milford, MA) was used to perform chromatographic separations. UPLC-MS raw data obtained in the negative mode were analyzed using the TargetLynx applications manager version 4.1 (Waters Corp., Milford, MA) to obtain the calibration equations and the quantitative concentration of each BA in the samples. The 12αOH BAs levels were the sum of CA, DCA, taurocholic acid (TCA), glycocholic acid (GCA), taurodeoxycholic acid (TDCA), and glycodeoxycholic acid (GDCA), and the levels of non-12αOH BAs included total the concentration of CDCA, LCA, UDCA, taurochenodeoxycholic acid (TCDCA), glycochenodeoxycholic acid (GCDCA), taurolithocholic acid (TLCA), glycolithocholic acid (GLCA), tauroursodeoxycholic acid (TUDCA), and glycoursodeoxycholic acid (GUDCA).

### 4.8. Statistical Analysis

Results are expressed as mean ± SD. Statistical significance was analyzed using the unpaired Student *t*-test. One-way analysis of variance (ANOVA) or Kruskal–Wallis tests were used to analyze the statistical significance in most parameters among the interventional groups. BA data were analyzed using the Mann–Whitney U test as most of the data had a nonnormal distribution. The software packages GraphPad Prism (GraphPad software 8.0, La Jolla, CA, USA) and SPSS (IBM SPSS version 22.0, Chicago, IL, USA) were used to statistically analyze the results. A *p*-value less than 0.05 was considered to be statistically significant.

## 5. Conclusions

To summarize, the intestinal microbiota plays a key role in regulating BA metabolism. In this study, we found that changes in BA signaling mediated by flaxseed regulating the gut microbiota altered host metabolism and contributed to NASH improvements. BAs are closely linked to the metabolic status, suggesting that regulation of BAs may be a promising strategy for the prevention of NASH.

## Figures and Tables

**Figure 1 ijms-22-10858-f001:**
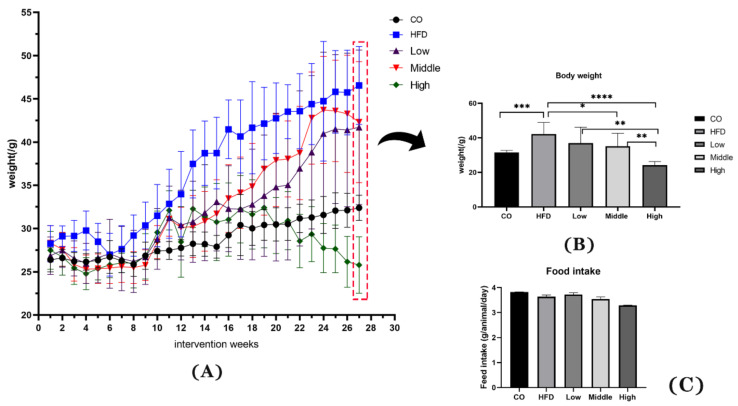
Effects of flaxseed powder on body weight (**A**) at different time points, (**B**) at week 28 of the intervention, and (**C**) the daily food intake between the five groups. Data are expressed as mean ± SD (n = 8 per group). * *p* < 0.05, ** *p* < 0.01, *** *p* < 0.001 and **** *p* < 0.001 (unpaired Student’s *t*-test); CO, control group; HFD, high-fat diet group; Low, 10% flaxseed powder intervention; Middle, 20% flaxseed powder intervention; High, 30% flaxseed powder intervention.

**Figure 2 ijms-22-10858-f002:**
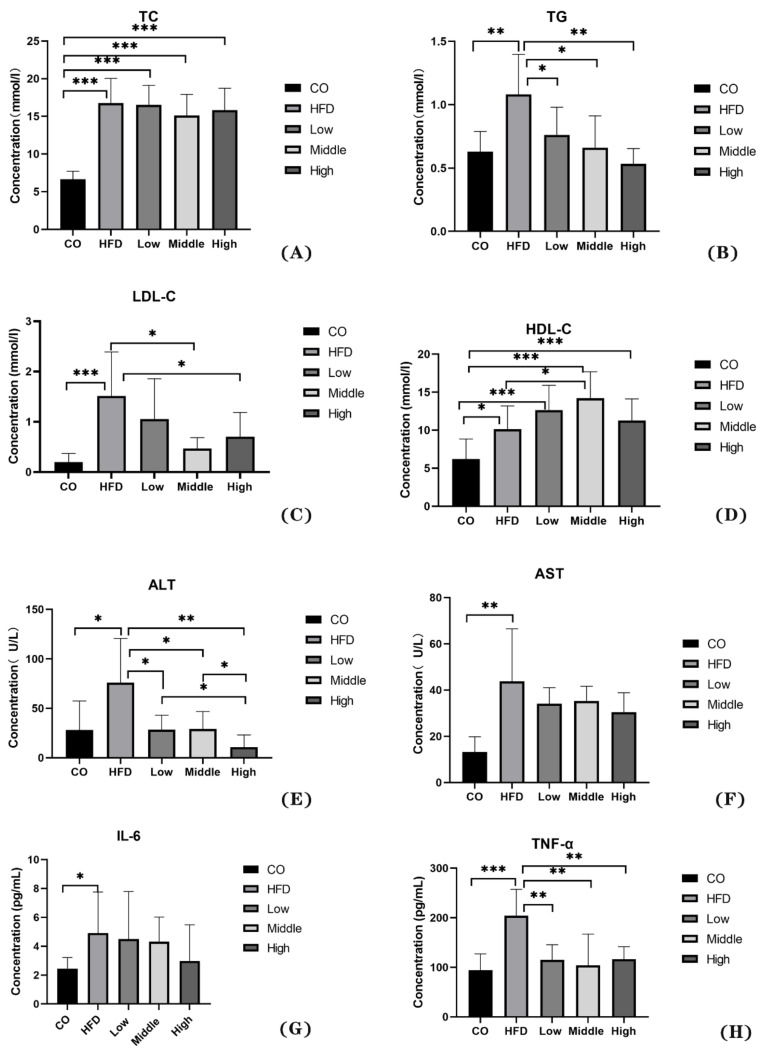
The physiological changes in the control, HFD, and FLA-fed groups: (**A**) TC, (**B**) TG, (**C**) LDL-C, (**D**) HDL-C, (**E**) ALT, (**F**) AST, (**G**) IL-6, and (**H**) TNF-α. Data are expressed as mean ± SD (n = 8 per group). * *p* < 0.05, ** *p* < 0.01, and *** *p* < 0.001 (unpaired Student’s *t*-test).

**Figure 3 ijms-22-10858-f003:**
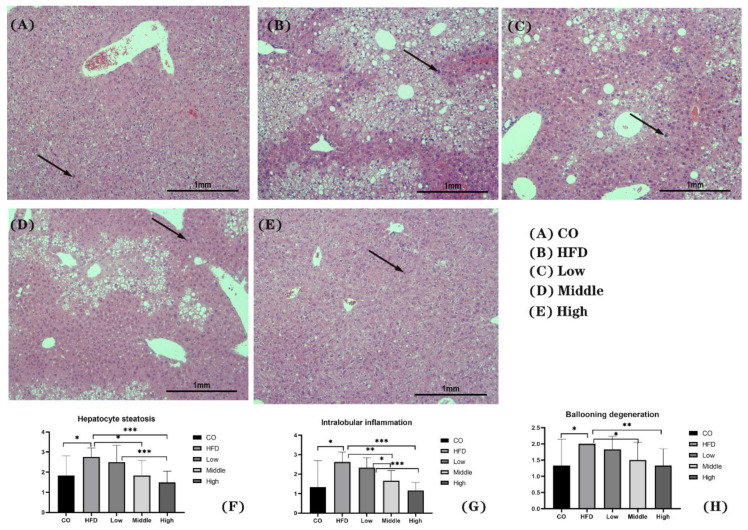
Representative images of H&E staining of liver sections (**A**–**E**) and score of hepatocyte steatosis (**F**), intralobular inflammation (**G**), and ballooning degeneration (**H**). Scale bars, 100 μm. Arrows indicate inflammatory factors. * *p* < 0.05, ** *p* < 0.01, and *** *p* < 0.001.

**Figure 4 ijms-22-10858-f004:**
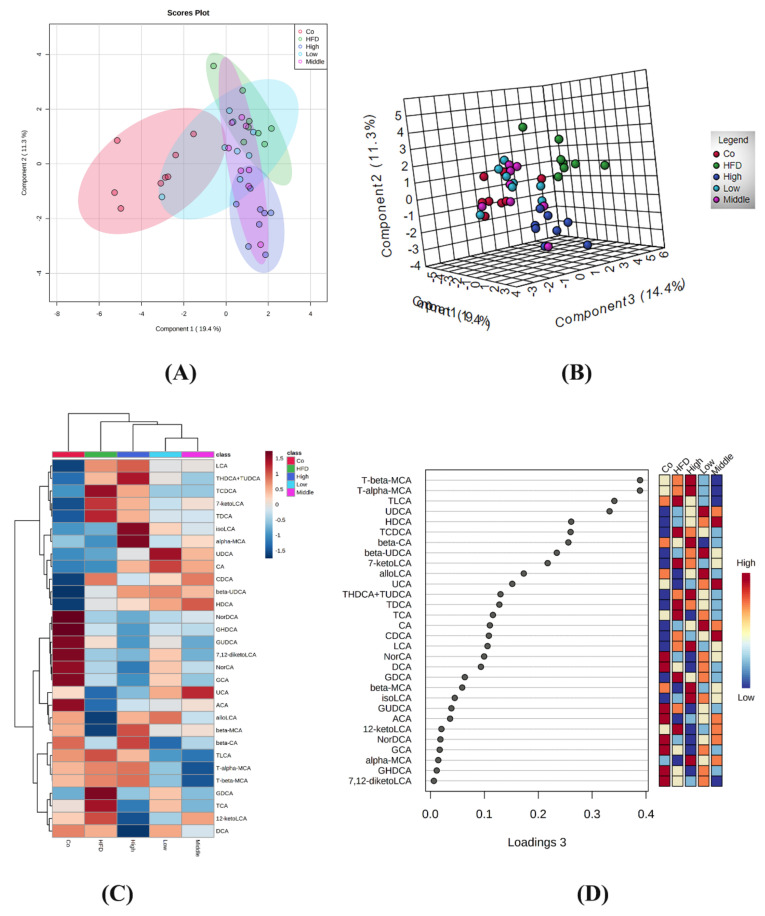
Different fecal BA profiles in the control, HFD, and FLA-fed groups. (**A**) Sparse partial least squares discriminant analysis (sPLS-DA) score plot of fecal BA profiles. This method can effectively reduce the complexity and enhance the interpretation ability of the model without reducing its prediction ability, in order to view the differences between groups to the greatest extent. (**B**) PCA score plot. The PCA score chart shows the aggregation and dispersion of samples. The closer the sample distribution points, the closer the composition and concentration of variables/molecules contained in these samples. (**C**) Heatmap of fecal BA profiles in the five groups (n = 40, 8 samples per group). (**D**) VIP scores of sPLS-DA based on the fecal BA profiles between the control, HFD, and FLA-fed groups. A bile acid with a VIP of more than 1 was considered important in the discrimination between the groups.

**Figure 5 ijms-22-10858-f005:**
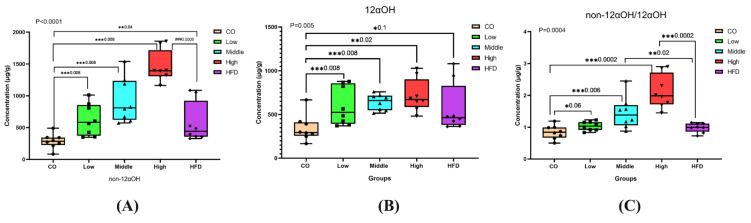
Differential fecal BAs with a relative expression of 12αOH and non-12αOH BA compositions in five groups. The data are presented as the mean ± SD. * *p* < 0.05, ** *p* < 0.01 and *** *p* < 0.001 (Mann–Whitney U test).

**Figure 6 ijms-22-10858-f006:**
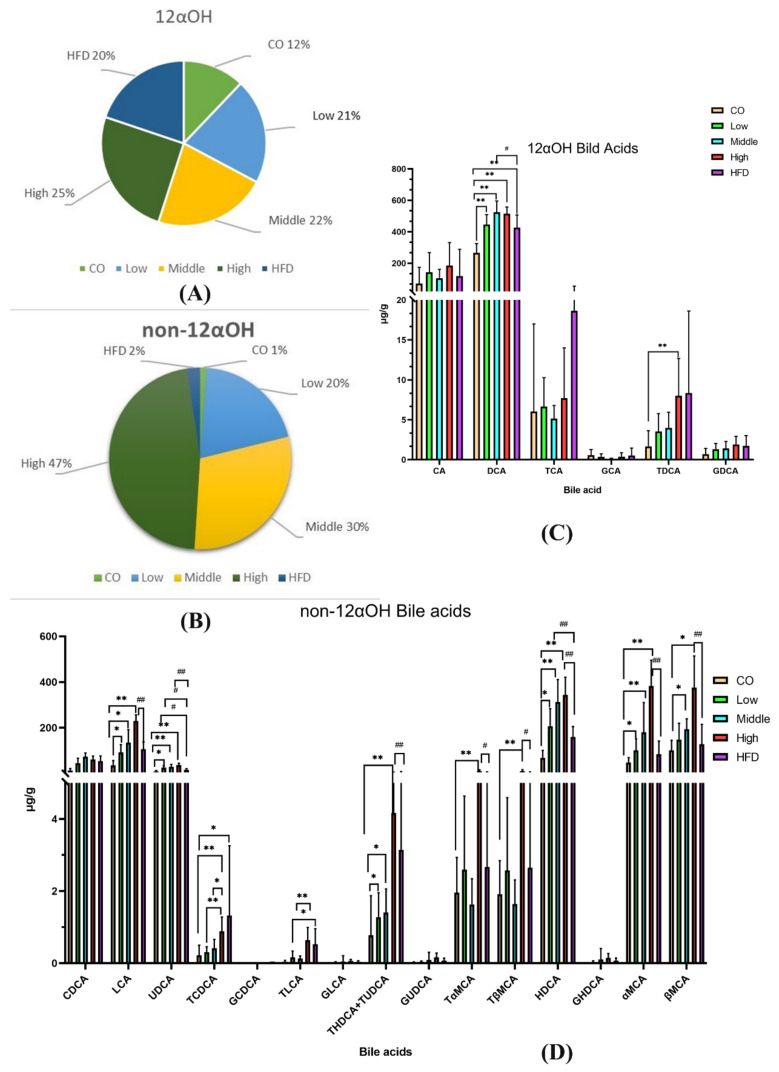
Dysregulated BA profiles in the HFD group and relative expansion of non-12αOH BA composition in the FLA-fed group. (**A**) The ratio of the 12αOH BA concentration of each group to the total content of the five groups. (**B**) The ratio of the non-12αOH BA concentration of each group to the total content of the five groups. (**C**) Comparison on 12αOH BA profiles in five groups. (**D**) Comparison on non-12αOH BA profiles in five groups. The data are presented as the mean ± SD. Compared with the control group, * *p* < 0.05 and ** *p* < 0.01 (Mann–Whitney U test); compared with the HFD group, # *p* < 0.05 and ## *p* < 0.01 (Mann–Whitney U test).

**Figure 7 ijms-22-10858-f007:**
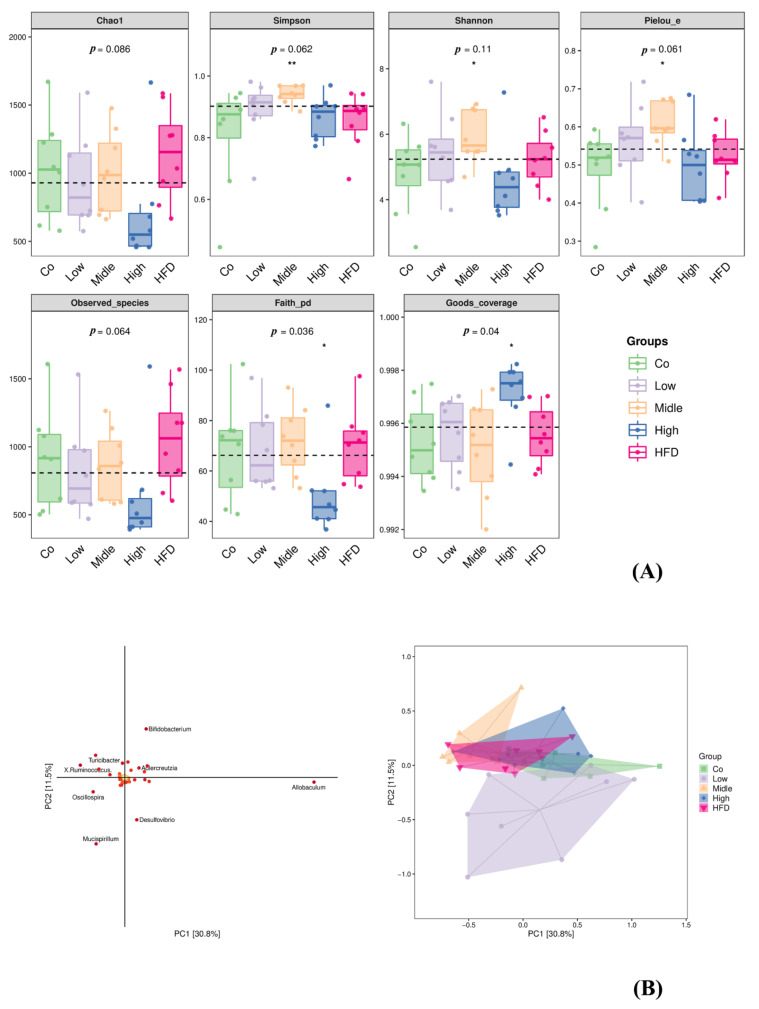
Box plot of alpha diversity index (**A**) and orthogonal partial least squares discriminant analysis (OPLS-DA) at genus level identified by metagenomic sequencing (**B**). α-Diversity of gut microbiota including Simpson index, Chao1, observed species, Faith’s PD, Pielou’s evenness, Good’s coverage, and Shannon index. Each panel corresponds to an alpha diversity index, which is identified in the gray area at the top. In each panel, the abscissa is the grouping label, and the ordinate is the value of the corresponding alpha diversity index. In the box line diagram, the meanings of each symbol are as follows: the upper and lower lines of the box, the interquartile range (IQR); the median line, the median; the upper and lower edges, the maximum and minimum inner circumferences (1.5 times IQR); the point on the outside of the upper and lower edges, the abnormal value. The number under the diversity index tag is the *p*-value of the Kruskal–Wallis test. Co, control group; HFD, high-fat diet; Low, 10% flaxseed group; Middle, 20% flaxseed group; High, 30% flaxseed group. * *p* < 0.05, ** *p* < 0.01.

**Figure 8 ijms-22-10858-f008:**
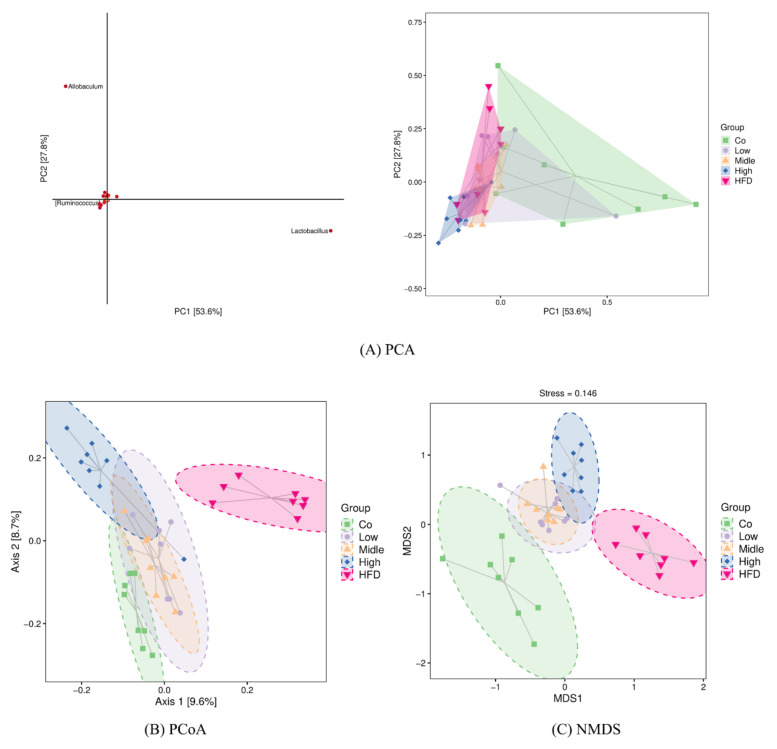
β-Diversity analysis using PCA (**A**), PCoA (**B**), and NMDS (**C**) based on unweighted Unifrac distance or Jaccard. Each point in the figure represents a sample, and points of different colors indicate different samples (groups). (**A**) The closer the projection distance of the two points on the coordinate axis, the more similar the species abundance composition between the two samples in the corresponding dimensions. (**B**) The percentages in the brackets of the coordinate axis represent the proportion of sample difference data (distance matrix) that can be explained by the corresponding coordinate axis. The closer the projection distance of the two points on the coordinate axis, the more similar the community composition of the two samples in the corresponding dimensions. (**C**) It is approximately believed that the closer (farther) the distance between the two points, the smaller (greater) the difference in microbial communities between the two samples. Co, control group; HFD, high-fat diet; Low, 10% flaxseed group; Middle, 20% flaxseed group; High, 30% flaxseed group.

**Figure 9 ijms-22-10858-f009:**
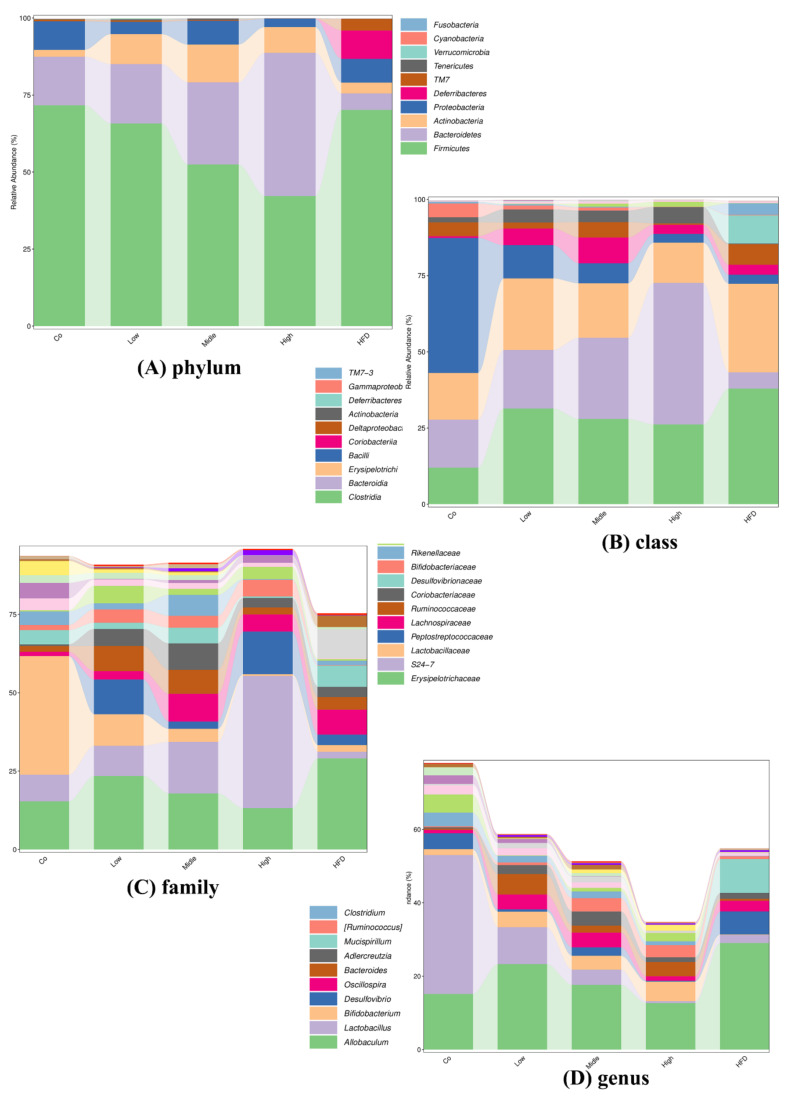
Relative abundance of the top 10 (**A**) phyla, (**B**) classes, (**C**) families, and (**D**) genera in each group.

**Figure 10 ijms-22-10858-f010:**
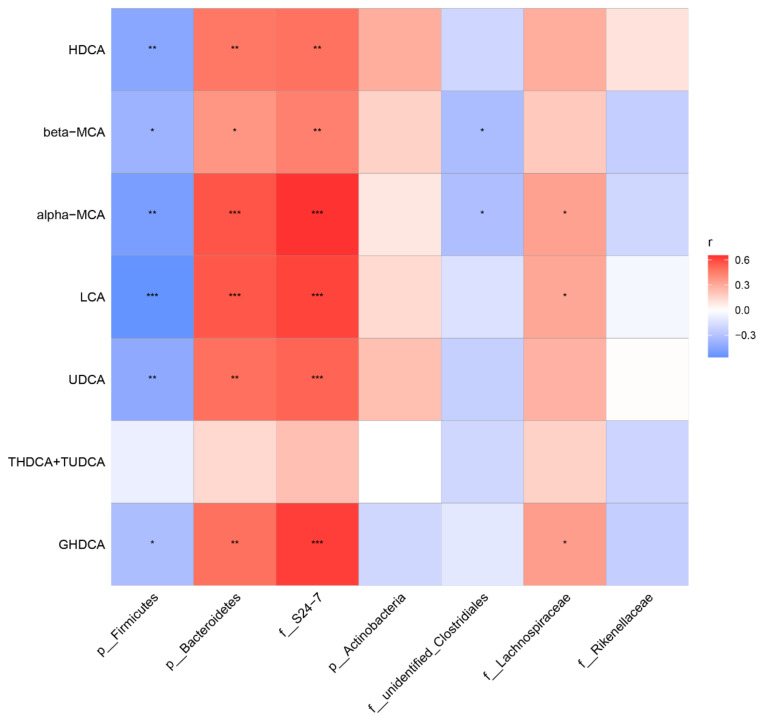
Heatmap of Spearman correlation coefficients between fecal BAs and microbiota from all samples in the five groups (n = 40, 8 samples per group). The gradient colors represent the correlation coefficients, with the red color being more positive, and the blue color indicating more negative. * *p* < 0.05, ** *p* < 0.01, *** *p* < 0.001.

**Figure 11 ijms-22-10858-f011:**
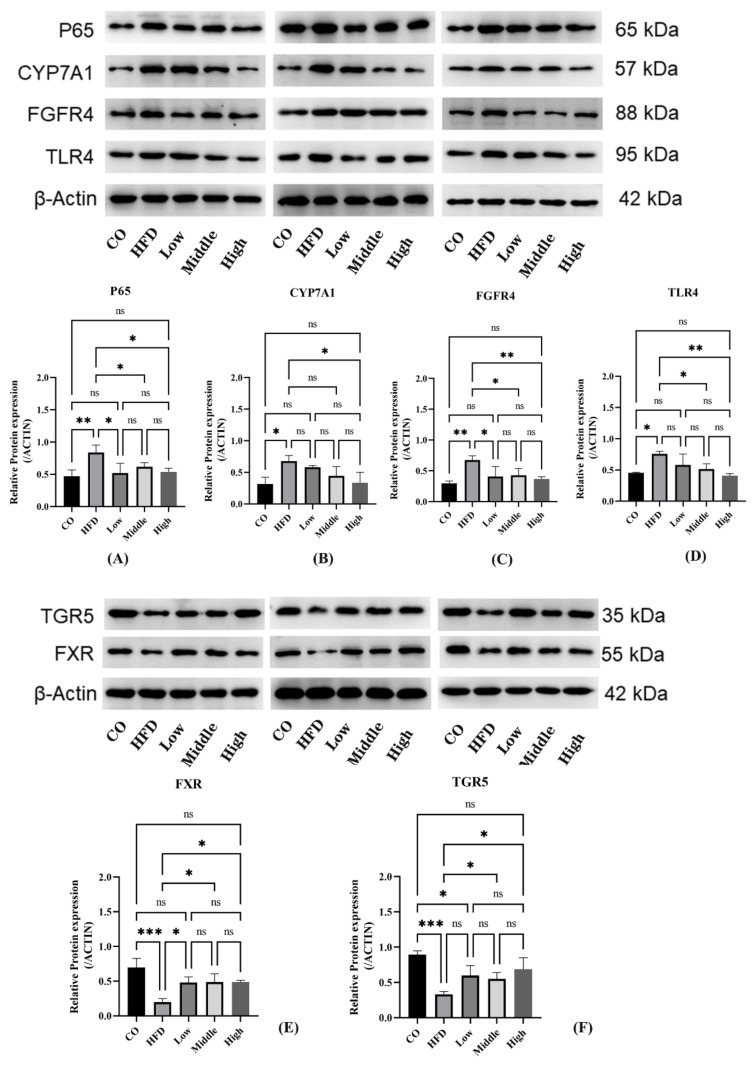
Analyses of mRNA expression and protein levels of key BA synthesis and inflammation factors in the liver and ileum. (**A**–**D**) The expression of CYP7A1, FGFR4, P65, TLR4, and CYP8B1 (n = 3 per group) in the liver was detected by Western blot. Data are presented as the mean ± SD. * *p* < 0.05 and ** *p* < 0.01 (unpaired Student’s *t*-test). (**E**,**F**) The expression of FXR and TGR5 in the ileum was detected by Western blot. Data are presented as the mean ± SD. * *p* < 0.05, ** *p* < 0.01, *** *p* < 0.001 (unpaired Student’s *t*-test).

## Data Availability

The data presented in this study are available in the graphs and tables provided in the manuscript.

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
