# Peer review of "Flaxseed Powder Attenuates Non-Alcoholic Steatohepatitis via Modulation of Gut Microbiota and Bile Acid Metabolism through Gut–Liver Axis"

_ijms, 2021, doi:10.3390/ijms221910858_

Round 1

Reviewer 1 Report

In a high-fat diet mice model, Yang et al. investigated that the flaxseed may prevent steatohepatitis via gut microbiota and bile acid regulation. This is my second time reviewing this article. Overall, this is a well-organized study to prove the study hypothesis. I have no further suggestions for this study.

Author Response

Comments and Suggestions for Authors

In a high-fat diet mice model, Yang et al. investigated that the flaxseed may prevent steatohepatitis via gut microbiota and bile acid regulation. This is my second time reviewing this article. Overall, this is a well-organized study to prove the study hypothesis. I have no further suggestions for this study.

Response to reviewer 1: Thank you very much for your affirmation of our research and manuscript. We look forward to the publication of the article as soon as possible. Thank you very much for your comments.

Reviewer 2 Report

The above article presents and discusses the potential NASH-preventing effect of flaxseed via regulation of gut microbiota composition and microbiota-related bile acids profile. From my perspective, the manuscript is too lengthy, which makes it challenging to find the essence. In addition, it contains many flaws, which are described below:

Introduction – the information provided that in the classic pathway of BA synthesis are generated CA and DCA and in alternate CDCA and LCA is an erroneous simplification. CDCA and its derivates (UDCA, LCA) is also produced in a classic pathway via CYP27A1 (see Pandak WM, Kakiyama G. The acidic pathway of bile acid synthesis: Not just an alternative pathway☆. Liver Res. 2019 May 21;3(2):88-98. doi: 10.1016/j.livres.2019.05.001.)

There is a lack of information in the methodology about the source of the flaxseed and its purity. Was the powder degreased? How does the addition of flaxseed change the initial % of energy from fat in the chow? It is not clear the percentage of flaxseed in the diet. Was the original high-fat diet modified to contain 10-20-30 % of flaxseed powder?

High-fat diet (HFD, 60% fat) for 14 weeks – it means 60 % energy from fat or 60 % of crude fat?

Surprisingly, after 14 weeks of HFD (60 % of fat), there is no difference in the body weight between control and HFD groups (see figure 1A, the start of flaxseed intervention). It is worth commenting.

There was no control group without HFD and with flaxseed supplementation. Therefore, there is no data on how flaxseed itself modulates gut microbiota and fecal BAs compostion in healthy animals.

It is worth adding a timeline showing the experimental design.

Figure 4 is not very informative. It is not clear what the results are presented here. The authors mentioned in the limitation of their study (Discussion) that they did not measure serum BAs. So what is figure 4c – “ Heatmap of spearman correlation coefficients between serum BAs and blood biochemical parameters from all samples in the three groups”? Moreover, primary, secondary as well as protective, and cytotoxic BAs should be distinguished. In its present way, that figure is hard to interpret. 

Moreover, the authors should inform in the introduction that murine and human BAs composition and microbiome profile are pretty different.

Figure 6 A and B – how were these pie charts constructed? Why are all the treatment groups combined to create 100 %? Shouldn't they show the percentage of 12aOH and non12aOH BAs relative to total BAs for each group separately?

Figures 7 and 8 should also be better described to be more informative for readers non-familiar with the calculation methodology used and the language describing it.

Figure 9A - due to the too-small font and species accumulation, the names of various phyla of bacteria (taxa) are virtually impossible to read. Should all of them be presented? Similarly, legends in figure 10 should be enlarged. Besides, in my opinion, figure 9 adds little to the manuscript and suggests its removal.

Overall, the manuscript should be significantly shortened and present more condensed and informative content. At its present form, it is not coherent, and it is hard to find the essential data within a bunch of different charts and analyses which are unclearly described.

The manuscript has not been prepared according to ijms template and needs an improvement in English writing. The text contains a lot of typos, stylistic and grammatical errors.

Round 2

Reviewer 2 Report

I appreciate the work the authors have done to address my concerns. However, there are still issues that remain to be resolved.

I keep suggesting modifying the paragraph concerning two pathways of BAs synthesis. Even in the reference, the authors cited (Wahlström A, et al. Intestinal Crosstalk between Bile Acids and Microbiota and Its Impact on Host Metabolism Cell Metab. 2016.) it is given: “Of note, whereas the alternative pathway predominantly generates CDCA, the classical pathway generates both CDCA and CA.” Therefore, the division proposed by the authors is an oversimplification.

I still encourage authors to include a brief description of the distinctions between human and murine microbiota and bile acids composition, especially since they compare their results to human studies (e.g., references 43,44,38, 46, etc.).

A lack of a control group on a standard diet supplemented with flaxseed is a considerable limitation of this study. 

Figure 4 - a legend contains a description of statistics: „*p<0•05 and **p<0•01 (Mann–Whitney U test).”. However, there is no indication of statistical significance on any of the charts. Moreover, as I suggested earlier, primary, secondary, and protective, and cytotoxic BAs should be distinguished. The way the data are presented makes it difficult to draw conclusions. 

Figure 6 - I don’t understand why the authors marked *p<0.1 as statistically significant (see figure legend). In Materials and Methods, they stated that: „A p-value less than 0.05 was considered to be statistically significant.”

There is a description entitled "Supplementary Materials" in the manuscript text regarding figures and tables from the supplement. However, the authors did not update it to the current version of the supplement (nine figures and six tables).

In my judgement, this study (particularly the number of figures) should be shortened considerably to present the most essential results.

In general, the fonts in graphs are too small, and some figures are therefore illegible (e.g., Figure 5, 7, 8, 9). 

This article could benefit from a review for grammar and spelling.

The manuscript has not been prepared according to ijms template, even the page numbering is missing. This makes it much more difficult for the reviewer to refer to specific fragments of the text. 

Round 3

Reviewer 2 Report

The authors addressed all issues that I raised. Answers are fine for the concerned points.

This manuscript is a resubmission of an earlier submission. The following is a list of the peer review reports and author responses from that submission.

Round 1

Reviewer 1 Report

The submitted manuscript concerns the effect of falxseed on non-alcoholic steatohepatitis after gut microbiota metabolism. The subject does not fit to the profile of International Journal of Molecular Science. In fact, the Authors had a wide range of results to present. A lot of them was included in the supplementary materials. In fact, the figures were missed in the main manuscript so the Authors did not show any results.

In general, the manuscript is very confusing, and not weel-organized. In addition, it was not written according to MDPI template. The results are missleading. The number of listed figures and references is exceeded, which makes the manuscript unclear. The Latin name of flaxseed (plant material) was not provided.

Reviewer 2 Report

In a high-fat diet mice model, Yang et al. investigated that the flaxseed may prevent steatohepatitis via gut microbiota and bile acid regulation. The study is interesting, but I still have some comments for this paper.

1. The main figures were not embedded in the main text. I will review the paper again after getting the main figures.

2. Several different alpha-diversity is calculated in the gut microbiome difference between mice model groups. Faith’s phylogenetic diversity is to measure the biodiversity that incorporates phylogenetic differences between species. Good’s coverage index estimates the percent of an entire species that are represented in a sample. Since Faith’s PD index and Good’s coverage index were significant differences between groups, the author should explain the diversity difference in these two indexes and the other five non-significant indexes.

3. A link between microbiota and bile acid, especially those bile acids presented a higher concentration in the high-dose flaxseed-fed group.